# Contribution of pks+ *Escherichia coli* (*E*. *coli*) to Colon Carcinogenesis

**DOI:** 10.3390/microorganisms12061111

**Published:** 2024-05-30

**Authors:** Mohammad Sadeghi, Denis Mestivier, Iradj Sobhani

**Affiliations:** 1EA7375–EC2M3: Early, Detection of Colonic Cancer by Using Microbial & Molecular Markers, Paris East Créteil University (UPEC), 94010 Créteil, France; denis.mestivier@u-pec.fr; 2Department of Gastroenterology, Assistance Publique–Hôpitaux de Paris (APHP), Henri Mondor Hospital, 94010 Créteil, France

**Keywords:** pks+ *E. coli*, colorectal cancer, colon, gut microbiota, diagnosis, therapy

## Abstract

Colorectal cancer (CRC) stands as a significant global health concern, ranking second in mortality and third in frequency among cancers worldwide. While only a small fraction of CRC cases can be attributed to inherited genetic mutations, the majority arise sporadically due to somatic mutations. Emerging evidence reveals gut microbiota dysbiosis to be a contributing factor, wherein polyketide synthase-positive *Escherichia coli* (pks+ *E. coli*) plays a pivotal role in CRC pathogenesis. pks+ bacteria produce colibactin, a genotoxic protein that causes deleterious effects on DNA within host colonocytes. In this review, we examine the role of the gut microbiota in colon carcinogenesis, elucidating how colibactin-producer bacteria induce DNA damage, promote genomic instability, disrupt the gut epithelial barrier, induce mucosal inflammation, modulate host immune responses, and influence cell cycle dynamics. Collectively, these actions foster a microenvironment conducive to tumor initiation and progression. Understanding the mechanisms underlying pks+ bacteria-mediated CRC development may pave the way for mass screening, early detection of tumors, and therapeutic strategies such as microbiota modulation, bacteria-targeted therapy, checkpoint inhibition of colibactin production and immunomodulatory pathways.

## 1. Introduction

Colorectal cancer (CRC) is one of the most common cancers, with over half of cases diagnosed between the ages of 50 and 75. Among cancers, CRC ranks second in mortality and third in frequency. In 2018, 1,849,000 new CRC cases and 881,000 CRC related-deaths were reported worldwide, representing approximately 10% of all diagnosed cancers [1]. 

It is estimated that CRC results from the accumulation of several gene mutations within a colonic stem cell. Less than 5% of CRCs can be explained by a constitutional mutation (such as Lynch syndrome or familial adenomatous polyposis (FAP)), while roughly 95% of CRCs are considered sporadic, meaning they result from the accumulation of somatic mutations and/or epigenetic events [2].

The composition of the microbiota performs common primary functions in all humans, although it is partly specific to each individual. Digestion of food by the intestinal microbiota is one of the essential metabolic functions of the human body. The abundance and diversity of bacteria both provide beneficial functions and insure a frontline defense against the colonization of so-called pathogenic or virulent bacteria.

Nearly 90% of fecal bacteria belong to two dominant phyla, Firmicutes and Bacteroides [3]. Dysbiosis is a change in the dominant clusters associated with a decrease in bacterial diversity in the microbiome (e.g., Bacteroides/Firmicutes ratio) and a decrease in bacterial density [4]. Dysbiosis has been characterized in metabolic diseases [5], chronic inflammatory bowel diseases (IBDs) such as Crohn’s disease and ulcerative colitis [6], neurological conditions (autism, multiple sclerosis, Alzheimer’s [7], Parkinson’s [8]) and CRC [9].

The first link between the bacterial population and colon carcinogenesis was reported in the 1970s through the identification of a relationship between *Streptococcus gallolyticus* (formerly *Streptococcus bovis* biotype 1) and CRC, with positive blood cultures in patients suffering from bacterial endocarditis in whom right-sided neoplasia was the source of the infection [10]. Since then, knowledge has expanded, implicating the microbiota as a potential actor in the genesis of CRC.

Moreover, in mouse models at a genetically high risk of cancer or subjected to a procarcinogenic chemical treatment (azoxymethane (AOM)), the presence of the intestinal microbiota acts as a significant co-stimulant. Mice raised in axenic (lacking intestinal microbiota) conditions exhibit a reduced incidence of pre-neoplastic lesions compared to those raised in non-axenic conditions [9,11]. Additionally, analysis of the 16S rRNA of the intestinal microbiota has revealed that the bacterial phylogenetic core was different in CRC patients than in individuals with normal colonoscopy findings, irrespective of age and gender [12].

One must distinguish between two types of bacteria linked to colon carcinogenesis [13]. The first type, presumed to initiate the carcinogenesis, likely by inducing DNA breaks and oxidative stress, are the so-called “driver” bacteria [14]. They consist of virulent pathogenic bacteria such as *Enterococcus fecalis*, *Escherichia coli*, and *Bacteroides fragilis*. The second type, known as “passenger” bacteria, are favored by changes in the tumor microenvironment and an increase in intestinal permeability leading to the expansion of the tumor. Their actions promote tumor growth and shape the tumor microenvironment. This group is mainly represented by *Fusobacterium* spp. and *Streptococcus gallolyticus* (formerly *Streptococcus bovis*, biotype 1) [13]. They can also induce DNA double-strand breaks in intestinal cells, leading to chromosomal instability, gene mutations and DNA epigenetic changes. Consequently, long-term interactions with luminal dysbiosis of the gut microbiota, including an imbalance in commensal bacteria populations, are considered to favor chronic DNA damage in host cells, akin to that observed in sporadic tumor tissues (e.g., deleterious mutations, epigenetic deviations) [15].

The direct causation of imprinted DNA changes resulting from a direct interaction between bacteria and host cells is not so far established. Here, we shift our focus to pkss, a group of bacteria proteins involved in various biological processes. These enzymes play a critical role in assembling complex organic molecules, such as fatty acids, antibiotics, and other biologically active substances. Their significance lies in their ability to catalyze a sequence of condensation reactions similar to cellular fatty acid synthesis, ultimately yielding a diverse range of compounds with unique chemical configurations.

The pks island, mainly prevalent among some *E. coli* strains in CRC patients, encodes various biosynthetic machinery, including no ribosomal peptide synthetases (NRPSs), polyketide synthases (pkss), colibactin syntheses as well as an efflux pump [16,17]. We focus on these bacteria because compelling data from preclinical studies and patient-based research support their role in CRC susceptibility or progression [18,19].

## 2. *E. coli*: A Versatile Bacterium

*E. coli* is a ubiquitous Gram-negative bacillus commonly found in the intestinal tract of humans and other mammals. While predominantly a commensal organism, certain strains have acquired virulence factors, which render them pathogenic. The versatility of *E. coli* in its roles as both a benign inhabitant and a pathogen has made it a subject of extensive study in microbiology [20].

### 2.1. Diverse Population and Phylogeny

Diverse populations of *E. coli* species inhabit the intestinal tract, with counts ranging from 107 to 109 colony-forming units per gram of feces. It is among the initial colonizers of the sterile intestines of newborns, acquired either from the mother during natural birth or from the environment during cesarean section [21]. Phylogenetically, *E. coli* is classified into seven main groups (A, B1, B2, C, D, E, and F) [22] based on the presence of virulence factors. While groups A and B1 are typically non-pathogenic, groups B2 and D are associated with both intestinal and extra-intestinal diseases [23].

### 2.2. Pathogenic Strains and Diseases

The intestinal pathotypes include Shiga toxin-producing *E. coli*, enteropathogenic *E. coli*, enterotoxigenic *E. coli*, enteroaggregative *E. coli*, and diffusely adherent *E. coli*, enteroinvasive *E. coli* (including Shigella strains), while the extra-intestinal pathotypes primarily include uropathogenic *E. coli* and strains responsible for neonatal meningitis. Adherence to enterocytes is a prerequisite step for pathogenic strains to initiate virulence pathways [20,24] (Table 1).

### 2.3. Environmental Adaptation

*E. coli* exhibits remarkable adaptability to various environments, including the host intestine and natural surroundings like water and soil [37]. The genomic makeup of *E. coli* is an adaptation to different environmental conditions. This occurs through mechanisms such as microbial genetic mutations and horizontal gene transfer. Factors influencing *E. coli* survival in the environment include temperature, nutrient availability, pH levels, and interactions with other microorganisms. Notably, certain strains of *E. coli* have been implicated in waterborne outbreaks, highlighting the potential health risks associated with environmental contamination.

### 2.4. Biofilm Formation and Resistance to Antibiotics

Biofilm formation around *E. coli* plays a significant role in its persistence in both environmental and host settings. Biofilms containing *E. coli* have been strongly associated with CRC, suggesting disease development [38,39] with a particular trend of strains toward antibiotic resistance; this also raised a hypothesis about the role of antibiotics in the increasing prevalence of CRC [40].

### 2.5. Protective Role of E. coli in Gut Microflora against Enteropathogens

*E. coli* is involved in a commensal relationship with its host. *E. coli EM0*, isolated from the feces of a healthy human volunteer, was shown to have a protective effect against the ETEC K88 strain in piglets. In germfree mice, the mechanism suggested was either “adaptation” of the EM0 strain inoculated first to germfree mice, as ultrastructural differences in the cell morphology were observed in vivo, or a slower generation time for the resistant enterobacteria mutant. Moreover, contributes to the development of the immune system in germfree mice enhance the production of cytokines in peritoneal and bone marrow macrophages and enhance the barrier effect against enteropathogens [41,42].

## 3. Role of *E. coli* in CRC Development

### 3.1. Epidemiological Evidence

*E. coli* is frequently found adherent to cancerous lesions and the nearby epithelium, often in large numbers, sometimes being the only culturable organisms in close contact with the tumor tissues [43].

In 1998, PCR tools detected *E. coli* in 60% of adenomas and 77% of CRC biopsy specimens compared to 12% in neighboring normal biopsies and 3% in normal control samples from healthy controls [44].

A French cohort study using culture tools found that 66% of CRC biopsies tested positive for *E. coli* compared to 19% in the diverticulosis control group. Among the virulent *E. coli* strains isolated from CRC tissue samples, 43% belonged to the B2 phylotype, a higher proportion than the 32% found in the diverticulosis control group. However, most *E. coli* in the CRC samples (57%) belonged to various other phylotypes [18].

The abundance of *E. coli* depends on the thickness of the mucus layer. When the mucus layer is removed, the normal colonic mucosa is relatively free of bacteria, whereas colon cancers and adjacent normal mucosa contain abundant flora, including *E. coli* [45].

Chronic inflammation of the intestinal tissues increases the risk of CRC. An epidemiological link between phylotype B2 *E. coli* and CRC suggests that Western diets, which cause chronic gut inflammation, may be a source of this risk. A longitudinal study in children indicated that B2 phylotype *E. coli* is a long-term colonizer of the human intestine, potentially producing prolonged inflammatory toxins, leading to malignancies like CRC [43].

### 3.2. Cyclomodulins: DNA-Damaging Compounds Produced by E. coli

*E. coli* produces compounds known as cyclomodulins, which introduce double-strand DNA breaks into target cells [46]. These include cytolethal-distending toxin (CDT), cytotoxic necrotizing factor (Cnf) [47], cycle-inhibiting factor (Cif) [48], intimin-dependent attachment encoded by eae [49] and colibactin [16] produced by the pks locus [50] (Table 2).

### 3.3. Cyclomodulins Promoting Cell Proliferation and Tissue Dedifferentiation

#### *E. coli* Cytotoxic Necrotizing Factor 1 (CNF1)

Cyclomodulin CNF is the only *E. coli* cyclomodulin that induces cell cycle progression [16,47]. It is close to three other cyclomodulins, dermonecrotic toxin (DNT) from *Bordetella* spp., Pasteurella Multocida Toxin (PMT) and CagA from *Helicobacter pylori*, that induce cell proliferation [46,52].

*E. coli* has three different types of this toxin, CNF1, CNF2 and CNF3, and a fourth type has been found in the bacterium *Yersinia enterocolitica*, named CNFy. These toxins have 65 to 90% common gene sequence similarity. Unlike CNF1, CNF3, and CNFy, which are encoded on the chromosome, CNF2 is situated on a transmissible plasmid [53].

This toxin activates Rho GTPases, blocking the cell cycle at the G2–M transition and DNA replication while inhibiting cytokinesis [54]. *E. coli* CNF1 binds to the tight junctions of host cells, which internalize it by endocytosis. It acts by deamidating a specific glutamine residue in the switch 2 domain that is crucial to GTP hydrolysis via glutamine 63 in Rho [55,56] or glutamine 61 in Rac and Cdc42 GTPases [57]. Hence, CNF1 may lock the G protein in the active state. This toxin activity confers new properties on the epithelial cells, including the cytoskeleton-dependent ability to behave as professional phagocytes, resulting in the formation of multinucleated cells [16] (Figure 1).

The potential contribution of CNF to carcinogenesis is multifaceted. Firstly, it alters Rho proteins similar to the changes observed in Ras’s proteins in various tumors [58,59]. Secondly, CNF-induced suppression of Rho GTPases activity, similar to *Salmonella* toxin’s impact on SopE and SptP factors, which correlates with gallbladder cancer development [60,61]. Thirdly, the similarity between the CNF1 and CagA proteins’ actions in *Helicobacter pylori*-infected gastric epithelial cells suggests CNF1’s long-term activation of NF-κB [62], known for its role in carcinogenesis [63]. This activation induces Cox2 proteins linked to chronic tissue inflammation and cancer [59,64].

Additionally, CNF promotes cell proliferation, inhibits apoptosis, and induces aneuploidy [62], while also disrupting cell attachment and motility via a RhoA-dependent mechanism, contributing to tissue dedifferentiation. These diverse effects underline CNF’s potential as a diagnostic marker and therapeutic target in understanding and addressing cancer progression [65,66].

**Figure 1 microorganisms-12-01111-f001:**
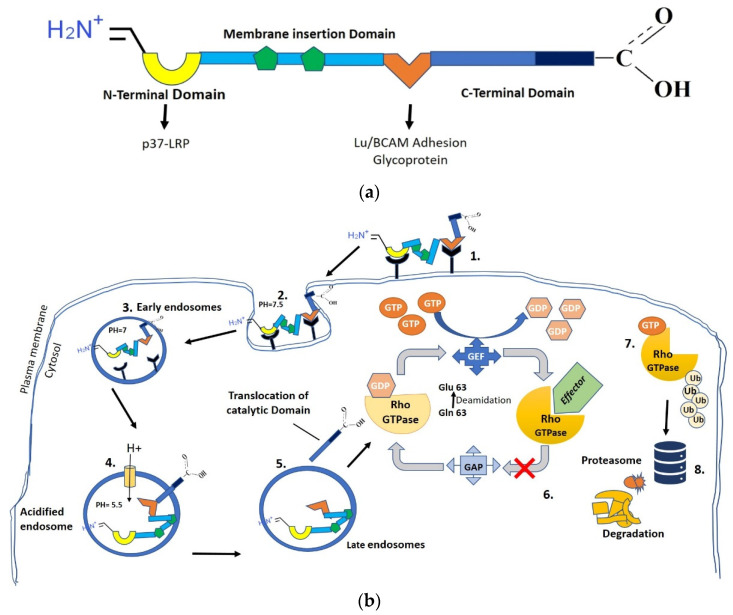
Structure and mechanism of CNF1 action. (**a**) Molecular structure of CNF1. The CNF1 molecule has two primary cell interaction sites: the N-terminus, which binds to p37/LRP, and the region directly adjacent to the catalytic domain, which serves as a high-affinity interaction site for Lu/BCAM (Lutheran (Lu) adhesion glycoprotein/basal cell adhesion molecule (BCAM)) [67]. (**b**) CNF1 is either directly secreted or associated with outer membrane vesicles. (1) Binds to two different cell receptors using their N-terminal and C-terminal domains. (2) Internalize into host cells through endocytosis and localize to early endosomes (3). Acidification of the late endosome triggers a conformational change (4), proteolytic processing (5), releasing the catalytic fragment into the cytosol (6). The fragment deamidates Rho GTPases at the cell membrane (7), initiating multiple cell responses, including ubiquitination and proteolytic degradation (8). (Modified figure based on Fabbri et al.’s (2013) and Tantillo et al.’s (2018) illustrations [47,68]).

### 3.4. Cyclomodulins Inhibiting Cellular Proliferation

#### 3.4.1. Cytolethal-Distending Toxin

Cytolethal-distending toxin (CDT) was identified in 1988 from strains of *E. coli* isolated from diarrheal patients’ feces [69]. CDTs are holotoxins comprising three subunits, CdtA, CdtB, and CdtC [51], whose molecular mass varies according to the bacterial species considered; they are coded by an operon, CdtA, CdtB, and CdtC genes [70].

This toxin is the first bacterial toxin shown to be involved in blocking the host cell cycle by arresting the transition between G2 and M. The cells exposed to CDTs have been shown to arrest in the G1 and/or G2 phases of the cell cycle or undergo apoptosis, depending on the cell type.

It acts like DNAses by inducing damage to the DNA of the infected cell. It induces DNA double-strand breaks, detectable with phosphorylation of histone H2AX and relocation of the DNA repair complex Mre11-Rad50 [51]. This damage will activate the DNA repair system, which causes cell cycle arrest in the G2/M phase [16]. Errors in repair may lead to a state of senescence if the infected cell is epithelial or mesenchymal [71], or death by apoptosis if the infected cell is part of the hematopoietic lineage, such as immune cells [72].

In addition, chronic exposure of cells to CDT induces chromosomal aberrations, resulting in an increase in the frequency of mutations, highlighting the genotoxic potential of CDT, which is favorable to cell transformation [73].

The molecular mechanisms triggered by CDT secretion from *E. coli* strain 1404, harboring the transferable plasmid pVir that encodes for the *E. coli* cytotoxic necrosis factor 2 (CNF2), have been extensively studied in HeLa co-cultures [74]. Within 4 h of interaction, HeLa cells exhibit the formation of giant mononuclear cells capable of blocking cells in the G2/M phase. Analysis of two subunits of the complex, cyclin B1 (the regulatory subunit) and protein kinase cdc2 (the catalytic unit), has revealed that CDT inhibits cdc2 kinase activity by preventing its dephosphorylation in normal cells. This inhibition pattern, shared among three partially related CDTs of *E. coli*, suggests that cells exposed to microbial CDT during the G2 and M phases are only blocked in the subsequent G2 phase [75].

These observations indicate that the toxin induces a cell arrest mechanism that initiates in the S phase, likely related to the DNA damage checkpoint system. This cell cycle arrest pattern resembles the checkpoint response to ionizing radiation, characterized by the activation of the ataxia telangiectasia-mutated (ATM) kinase and ATM-dependent induction of the tumor suppressor p53. Additionally, phosphorylation of histone H2AX and re-localization of DNA repair proteins such as Mre11 and Rad50 to sites of DNA double-strand breaks are observed [72,76,77].

Consequently, failure to repair this damage can lead to a mechanism termed “stress-induced premature senescence”. The entry into stress-induced premature senescence of damaged cells can result in uncontrolled proliferation, akin to tumor cells, irrespective of whether it is independent of soluble pro-inflammatory cytokines or not [78]. Although breaks in the double-stranded DNA of host cells and ATM kinase-mediated responses are observed in short-term experiments with mammalian cells [72,79], it remains unclear whether CdtB acts as a true DNase or indirectly induces DNA damage. The exact molecular mechanism by which CdtB induces DNA damage remains to be elucidated.

#### 3.4.2. Cell Cycle-Inhibiting Factor (Cif)

Cyclic inhibitory factor (Cif) is another protein secreted by enteropathogenic and enterohemorrhagic *E. coli* [78] with two distinct domains with cytopathic activity on Hela cells in vitro [80]: an N-terminal domain that carries a peptide for addressing the type III secretion system [81], which allows it to be injected into the cell, and a C-terminal domain that carries the enzymatic activity of the toxin. 

These cytopathic effects result from an irreversible cell cycle arrest at the G2/M transition, accumulation of cyclin-dependent kinase inhibitors (CKIs) p21waf1/cip1 and p27kip1, formation of actin stress fibers (cytoskeleton reorganization), and sustained inhibitory phosphorylation of the mitosis inducer, CDK1. The protein is a divergent member of a family of enzymes, including cysteine proteases and acetyltransferases, which have a conserved catalytic triad [81]. However, it is neither a genotoxic nor an activator of DNA damage checkpoint pathways [82]. Hence, Cif activates a DNA damage-independent signaling pathway that leads to the inhibition of the G2/M transition.

Inhibition of the ubiquitin/proteasome pathway can affect key proteins involved in CRC. The cellular consequences are the blocking of apoptosis. However, the cells can continue to synthesize DNA and the DNA content of the cells increases through endoreduplication to reach 16n, potentially leading to genetic abnormalities that may promote tumor development [46]. Inactivation of the Chk1 and 2 kinases does not alleviate Cif-induced inhibition of mitosis, suggesting that Cif initiates cell cycle arrest independently of the ATM-Chk2 and ATR-Chk1 checkpoint pathway [82]. This arrest is not a result of cytoskeleton alterations, as inhibition of the stress fiber formation by Rho inhibitors does not prevent the accumulation of G2-arrested cells.

### 3.5. Colibactin

In 2006, Nougayrède et al. identified colibactin for the first time in a strain of *E. coli* (IHE3034) isolated from neonatal meningitis. This toxin is encoded by the pks gene [16] and acts as a cyclomodulin, disrupting the eukaryotic cell cycle. The origin and prevalence of the colibactin island among enterobacteria remain unknown. Using isotopic labeling, DNA adductomics, and mass spectrometry, colibactin has been shown to contain double cyclopropane caps that form DNA cross-links by alkylating adenine residues [83,84].

#### 3.5.1. pks and Its Role in Synthesizing Various Compounds

pkss are vital enzymes engaged in the production of diverse compounds, including fatty acids, antibiotics, and other biologically active molecules. They are pivotal in assembling intricate organic compounds involved in cellular fatty acid synthesis. Although pks enzymes are present in various microorganisms, such as bacteria, fungi, plants, and specific marine invertebrates, contributing to the biosynthesis, certain *E. coli* strains carry a biosynthetic gene cluster (BGC) known as pks that encodes colibactin [85].

Ordinarily, pks enzymes enable the repetitive condensation of building blocks such as acetyl-CoA and malonyl-CoA to generate a linear polyketide chain. This chain undergoes structural modifications such as cyclization, reduction, and oxidation [86].

Three different types are described according to the structural and mechanistic attributes regarding various pks enzymes. pkss of type I are characterized by multiple catalytic domains within a single polypeptide chain, facilitating a specific step in polyketide chain elongation and modification [87].

pkss of type II are characterized by separate, monofunctional enzymes catalyzing a single reaction, transferring intermediates from one enzyme to another, all ranged along a biosynthetic pathway [88].

pkss of type III are characterized by their small size and homodimeric structures, with iterative condensation of the basic starter units for generating polyketides [89].

The genomic island of colibactin comprises 20 open reading frames (ORFs), with 8 coding for putative pks, non-ribosomal peptide synthetases, and their hybrid forms. Prior to the discovery of this island, the only known non-ribosomal peptide and polyketide/non-ribosomal peptide hybrids in Enterobacteriaceae were the iron chelators enterobactin and yersiniabactin, respectively [90,91]. In contrast to these iron chelators, the synthesized hybrid non-ribosomal peptide-polyketide colibactin induces a cytopathic effect on eukaryotic cells in vitro. Cocultivation of colibactin island-positive bacteria with eukaryotic cells leads to the induction of DNA double-strand breaks, causing cells to arrest in the G2 phase of the cell cycle and exhibit megalocytosis and cell death [16]. These effects resemble those of the cyclomodulin cytolethal-distending toxin, although the biological function of colibactin in vivo remains unknown.

The pks genomic island includes 19 genes that code for the complex enzymatic machinery responsible for the synthesis of colibactin (Table 3, Figure 2). It includes 54 kb of a chromosomal region located in the asnW tRNA locus, a site of DNA insertion and exchange [36,92].

Currently, 16 of them are known and essential for obtaining a colibactin effect. They include polyketide synthases (pkss) such as clbC, clbI, and clbO, non-ribosomal peptide synthases (NRPSs) such as clbH, clbJ, and clbN, and non-ribosomal polyketide-peptide synthases (pksNRPSs) such as clbB and clbK.

The pks enzyme would participate in the maturation of the compound in the periplasmic compartment [96].

**Figure 2 microorganisms-12-01111-f002:**
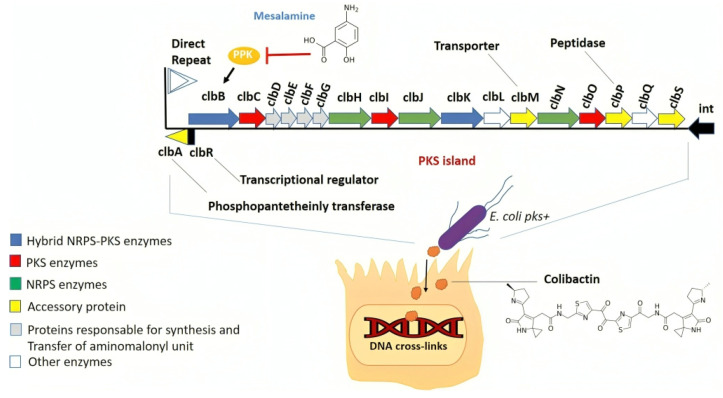
Organization of the pks island. pks island encodes the genes required for the synthesis of colibactin. Genes coding for pks (red), non-ribosomal peptide synthase genes (green) and the two fusion genes (blue) are shown. They are in connection with the regulator clbR and the integrase (int) (black). Protein functions for ClbA, ClbM, ClbP and ClbS are indicated. Polyphosphate kinase (PPK) activity is essential for ClbB function and colibactin metabolism. Mesalamine, a medication for ulcerative colitis, reduces PPK activity and colibactin production. (Modified figure based on Nougayrède et al.’s (2006) [16], and Faïs et al.’s (2018) illustrations [96]).

NRPSs and pkss are very large synthases (megasynthases) composed of several subdomains and are categorized into three types on their enzymatic activity. These activities include thiolation domains (T, thiolation; ACP, Acyl Carrier Protein; PCP, Peptide Carrier Protein) where a thioester bond is formed between the amino acid and the mega synthase; acyltransferase domains for pkss; and adenylation domains for NRPSs. These bonds facilitate the attachment of the thiolation domain. Additionally, condensation domains (ketoacyl synthase on the pks; condensation for NRPS) create a peptide bond between two monomers and condense the added monomer onto the elongating chain [97]. Although the complete structure of colibactin is still unknown, several studies published in recent years have identified numerous synthetic intermediates [98,99].

First, the biosynthesis of colibactin requires the activation of pks/NRPS megasynthases (non-ribosomal peptides) by phosphopantetheinyl transferase ClbA. Once activated, megasynthases recruit their specific monomers. The first enzyme involved in the synthesis is ClbN, whose substrate is asparagine. The assembly then proceeds with the successful intervention of the ClbB-ClbC-ClbH-ClbI-ClbJ-ClbK enzymes, which utilize various common substrates such as malonyl-CoA, alanine, serine, glycine, cysteine or alternative, unusual substrates such as aminocyclopropane-carboxylic acid to yield the toxin in its pre-colibactin form. Inactive pre-colibactin is exported into the periplasm via ClbM, which acts as an efflux pump [100]. ClbM is inserted into the inner membrane. The pre-colibactin is then hydrolyzed, resulting in the separation of the cleavage product, or N-myristoyl-D-asparagine, and the active colibactin (ClbP), which is anchored to the inner membrane and possesses D-aminopeptidase activity [101]. The inactive pre-colibactin (Figure 3) could be a protective mechanism for the bacterium against the genotoxic compounds it produces. In 2016, an alternative protective mechanism was described after the ClbS protein was identified. The latter may protect the colibactin-producing bacterium by sequestration or modification of the colibactin molecules present in the cytoplasm [102].

Multiple studies have shown that pks gene expression is influenced by factors such as inflammation [103,104], iron availability [105,106], microbial metabolites [107], oxygen availability [108], and dietary oligosaccharides [109].

Epidemiological evidence shows a higher prevalence of pks+ *E. coli* in CRC patients compared to those with inflammatory bowel disease (IBD) or healthy controls [19,110,111].

Five pks islet genes (ClbG, ClbH, ClbL, ClbM, and ClbS) have been shown to be activated during tumor development in an inflammation-dependent manner, suggesting that various colibactin syntheses intervene in the microenvironment during CRC development. In the context of chronic inflammation such as ulcerative colitis, mesalamine and anti-inflammatory medicine reduces PPK activity and colibactin production [108]. One could speculate that reducing mucosal inflammation is a way to decrease the cancer risk through colibactin diminution in eukaryotic host cells [18,83].

#### 3.5.2. Colibactin-Producing *E. coli* and CRC Mutations

A recent study by Pleguezuelos-Manzano et al. utilized human intestinal organoids to demonstrate that pks+ *E. coli* induce a mutational signature associated with CRC [112]. Chronic exposure of human organoids to pks+ *E. coli* but not clbQ-deficient *E. coli* (which cannot synthesize colibactin) resulted in unique transcriptional signatures characterized by single-base substitutions (SBSs) and insertion-deletions (IDs) within AT-rich DNA motifs. Whole-genome sequencing (WGS) revealed a distinct pattern of somatic mutations, which includes thymine insertions at thymine homopolymers and increased thymine to other base substitutions, particularly in adenine-rich regions, consistent with colibactin’s mutagenic activity [83].

These mutagenic signatures are present in approximately 5% of metastatic tumors, primarily from CRC-derived primary sites [112].

Notably, this signature was found in 112 known CRC driver mutations, with APC (adenomatous polyposis coli) showing the highest number of matching mutations. Additional studies have also identified colibactin-specific mutations in large surveys of CRC genomes and in genes related to p53 signaling. Colibactin-related mutations were also found in healthy human colon biopsies, suggesting that early-life exposure to colibactin might increase the CRC risk later [113]. Approximately 20% of healthy individuals harbor pks+ *E. coli*, raising questions about why only some develop CRC [18,114].

Systematic studies have shown that 14 out of the 19 genes encoded by the pks island are required for its genotoxic effects [82]. While the colibactin biosynthesis pathway is well-characterized and pks gene expression is upregulated in CRC models and patients, regulatory factors remain understudied [17,115]. Experiments such as profiling conditioned media from clbQ-deficient and -sufficient *E. coli* in mutation assays may identify bioactive compounds modulating colibactin’s effects. Targeting pks-encoded products like ClbM and ClbP reduces genotoxicity and the tumor burden, and inhibiting polyphosphate kinase (PPK) with mesalamine reduces colibactin production.

Overall, the combination of basic, clinical, and bioinformatic approaches has revealed that colibactin directly drives CRC-associated mutations. Determining the optimal strategies for intervention with pks+ *E. coli* remains a critical question, but colibactin inhibitors represent a promising treatment avenue for CRC prevention [84].

#### 3.5.3. Activation of Colibactin Biosynthesis Pathway

Once the colibactin biosynthesis genes are acquired, their expression is typically regulated by environmental signals or specific transcriptional factors within the bacterial cell. Upon sensing appropriate stimuli, such as host-derived metabolites or changes in the microenvironmental conditions, the regulatory elements controlling the colibactin biosynthesis pathway are activated. This activation results in the production of colibactin and designs the pks+ phenotype [98] of the strain.

### 3.6. Effects of Colibactin

#### 3.6.1. Genotoxic Effects

Colibactin alkylates DNA, leading to the formation of interstrand crosslinks and DNA double-strand breaks, which can result in genomic instability and mutations in critical oncogenes or tumor suppressor genes. The accumulation of DNA damage promotes the initiation and progression of CRC, contributing to the transformation of normal colonic epithelial cells into malignant tumor cells [112,116].

One of the markers most commonly used to assess DNA damage and more particularly DNA double-strand breaks is the phosphorylated form of serine 139 in histone H2AX [117]. The γH2AX signal of infected cells increases according to the infection dose, as assessed by the “comet test” [118], amplifying the toxic effect.

**Figure 3 microorganisms-12-01111-f003:**
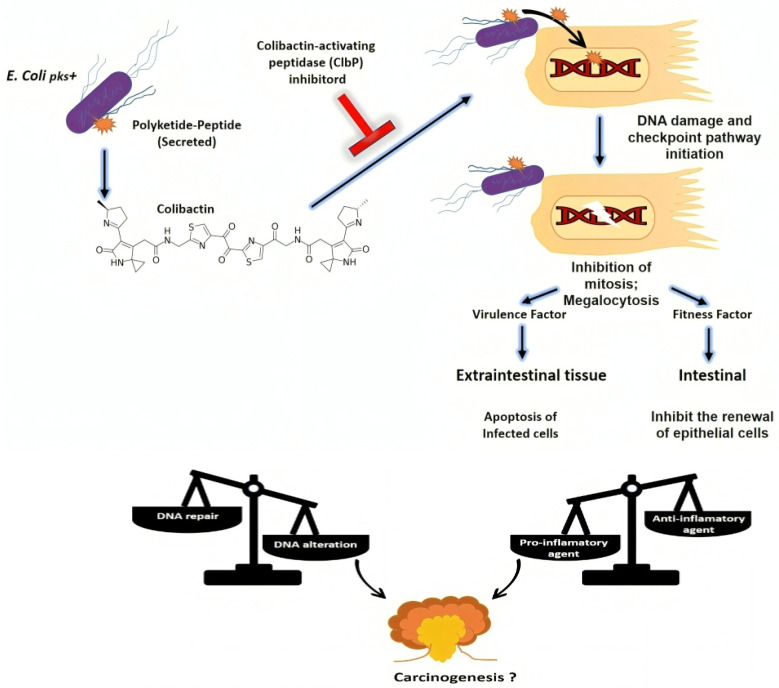
Role of colibactin and impact on the host. Repeated exposure of enterocytes to pks+ bacteria could result in the chronic formation of DNA double-strand breaks. Inhibitors of colibactin-activating ClbP peptidase biosynthesis block the genotoxic effects of colibactin on eukaryotic cells [119].

#### 3.6.2. Induction of Genomic Instability

pks+ bacteria contribute to cancer progression by inducing genomic instability in colorectal tumors. Colibactin-induced DNA damage not only initiates carcinogenesis but also promotes genomic instability by impairing DNA repair mechanisms and increasing the frequency of mutagenic events [117] and facilitating the emergence of aggressive tumor subclones with enhanced proliferative and invasive capacities. 

#### 3.6.3. Disruption of Gut Epithelial Barrier and Induction of Inflammation

pks+ bacteria can disrupt the integrity of the gut epithelial barrier, facilitating bacterial translocation and interaction with host cells. Colibactin-induced DNA damage may compromise the function of colonic epithelial cells, leading to the increased permeability of the intestinal mucosa and leakage of bacterial toxins and antigens into the underlying tissues. This breach of the epithelial barrier allows pks+ bacteria to directly interact with immune cells and promote inflammatory responses, contributing to CRC [120]. pks+ bacteria, through the production of colibactin and other virulence factors, can induce chronic inflammation in the colon, enhancing the risk of CRC.

Studies have shown that inflammation alters host physiology to promote cancer. By examining the gut microbiota as a target of inflammation influencing CRC progression, high-throughput sequencing revealed that inflammation alters the gut microbial composition in colitis-susceptible interleukin-10-deficient mice [114]. Inflammatory signaling pathways activated in response to colibactin exposure contribute to the recruitment of immune cells, release of pro-inflammatory cytokines, and generation of reactive oxygen species (ROS). 

#### 3.6.4. Modulation of Host Immune Responses

pks+ bacteria can modulate host immune responses in the colon, which may induce regulatory T-cell differentiation and myeloid-derived suppressor cell recruitment, further dampening the anti-tumor immune responses and facilitating tumor progression. Colibactin-induced DNA damage and inflammation may alter the function of immune cells within the gut-associated lymphoid tissue, impairing anti-tumor immune surveillance and promoting immune evasion by tumor cells.

Single colonization with a commensal *E. coli* strain causes invasive cancer in azoxymethane (AOM)-treated Il10 mice, and deletion of the genotoxic polyketide synthase (pks) island from this strain reduces tumor multiplicity and invasiveness in AOM/Il10−/− mice intestinal inflammation without change [114].

Mucosal-associated pks+ *E. coli* have been found in a high percentage of patients with inflammatory bowel disease (IBD) and CRC. 

#### 3.6.5. Cell Cycle Effects

A study that investigated the cell cycle of cells infected with pks+ bacteria showed that cells with megalocytosis (an abnormally large cell size) also exhibit cell cycle arrest at the G2/M transition with synchronized cells in the G1/S phase, showing that infected cells stay longer in the S phase of the cycle and stop at the G2/M transition [16]. Colibactin-induced cell senescence is characterized by irreversible cell cycle arrest associated with significant morphological and physiological changes. Following infection by pks+ *E.coli*, senescent cells exhibit megalocytosis, lysosomal β-galactosidase activity at pH 6, the accumulation of p53 and p21 proteins involved in cell cycle arrest, as well as a decrease in SENP1 peptidase [115], although senescence can have an anti- or pro-tumor role depending on the context [121]. It has been proposed that if the bacteria/cell ratio is high, the bacteria may exert anti-tumor activity by inducing senescence in the majority of cancer cells through cell cycle blockade, while a low ratio may induce senescence in a limited number of cancer cells; this suggests that the bacteria could have a pro-tumor paracrine activity thanks to the secretory phenotype associated with senescence [115].

Colibactin has also been shown to inhibit the activation of cyclin-dependent kinase 1 (Cdk1), a protein involved in cell cycle regulation, which could lead to cell death or the formation of cells with abnormal genomic content [122,123,124].

#### 3.6.6. β-Glucuronidase Activity

Research indicates that individuals at high risk of CRC and those diagnosed with CRC exhibit a 1.7-fold increase in β-glucuronidase activity compared to healthy counterparts [125]. This enzyme, β-glucuronidase, is implicated in CRC by aiding in the synthesis of carcinogenic metabolites. Normally, the liver metabolizes Dimethylhydrazine, a colon carcinogen, with small amounts of its procarcinogenic metabolite, methylazoxymethanol, excreted in the bile and released into the large intestine. However, the β-glucuronidase activity of *E. coli* can hydrolyze Dimethylhydrazine, releasing methylazoxymethanol in the colon [125]. Studies show that axenic animals treated with Dimethylhydrazine develop fewer colon tumors compared to conventional animals. Furthermore, administering a β-glucuronidase inhibitor in rats treated with Dimethylhydrazine results in a decrease in tumor formation. These findings underscore the potential of β-glucuronidase as a key player in CRC development, highlighting avenues for therapeutic intervention and preventative measures. The effects of colibactin are summarized in Table 4.

#### 3.6.7. Potential Presence of pks+ in Normal Colon Tissue of Cancer Patients

A study comparing the genomes of distant normal crypts, normal crypts adjacent to tumors, and cancerous glands from the same patients showed that pks+ *E. coli* is widespread in the normal colon of cancer patients and may be a candidate process responsible for multiple mutations in cancer-promoting genes in malignancies [126].

#### 3.6.8. Heterogeneity of pks+ Bacterial Localization

The localization of pks+ bacteria within colon tissues may exhibit heterogeneity among individuals and across different stages of CRC. While some patients may show a predominance of pks+ bacteria within tumor tissues, others may exhibit a more diffuse distribution, with bacteria present in both cancerous and healthy regions of the colon. Factors such as host immune responses, microbial interactions, and environmental influences may influence the spatial distribution of pks+ bacteria within colon tissues. 

The results of a study showed that pks+ *E. coli* is more prevalent in stage 0–1 tumor tissue compared to normal mucosal tissue or stage II–IV tumor tissue. A high abundance of pks+ *E. coli* in tumor tissue was significantly associated with a shallower tumor depth and absence of lymph node metastasis in multivariate logistic analyses. The pks+ *E. coli*-low and -negative groups were significantly associated with shorter CRC-specific survival and recurrence-free survival compared to the pks+ *E. coli*-high group [127].

However, to date, there are no studies on the causal relationship between pks+ signatures and driver mutations and the linear relationship between pks+ and incidence of CRC is largely unclear [126].

## 4. Bacteria Associated with pks+

The identification of bacterial species associated with pks+ production involves comprehensive microbial profiling and genetic analysis to determine the presence of pks gene clusters and colibactin biosynthesis pathways. 

While not all *E. coli* strains carry the pks island, those that do have been implicated in CRC and are of particular interest in studies investigating the role of pks+ bacteria in cancer initiation and progression [18]. Other bacterial species have been shown pks production, including certain strains of *Klebsiella pneumoniae*, *Enterobacter aerogenes* and *Citrobacter koseri* [128], suggesting that these bacteria in the gut microbiota may exert similar effects in colon carcinogenesis.

### Colibactin Induces Prophages

The effect of colibactin on the induction of prophages in neighboring bacteria has been reported in co-culture procedures and genetic analyses, including various species of bacteria in the human gut. Consequently, a protein called ClbS and several ClbS-like protein-related bacteria have been characterized as colibactin-induced DNA damage and prophage [129].

## 5. Prevention and Treatment Strategies

While pks+ *E. coli* is associated with CRC, its exact role in CRC prevention is not well defined. However, maintaining a balanced gut microbiota through dietary interventions, probiotics, or other means may indirectly reduce the abundance or activity of pks+ *E. coli* and potentially lower the CRC risk, enabling novel therapeutics targets [108,119].

### 5.1. Inhibiting pks Biosynthesis

The aerobic respiration control (ArcA) positively regulates colibactin production and the genotoxicity of pks+ *E. coli* in response to oxygen availability. Thus, exposure to oxygen can inhibit colibactin synthesis [130]. One would speculate that modulating the oxygen levels in the gut may offer a potential tool for preventing or treating colon cancer associated with colibactin-producing bacteria [130].

Alternatively, inhibitors targeting the pks machinery, such as small molecules or peptides mimicking pre-colibactin that inhibit colibactin-activating peptidase ClbP (Figure 3), may be used to inhibit colibactin synthesis [119,131,132]. These peptides can be designed to target specific regions of the pks involved in catalysis, substrate binding, or protein–protein interactions, thereby disrupting pks function [133].

SiRNA molecules designed to target and silence pks gene expression act by stimulating the degradation of mRNA molecules that encode pks enzymes, thereby reducing the synthesis of functional pks proteins and inhibiting their activity.

### 5.2. Microbiota Modulation

Strategies aimed at modulating the gut microbiota composition, likely through dietary interventions, probiotics, prebiotics, and fecal microbiota transplantation (FMT), may help to reduce the proliferation of pks+ bacteria [134,135]. Alternatively, in high-risk individuals or patients with CRC predisposition syndromes and presence of pks+ bacteria, antibiotics may be used.

### 5.3. Phage Therapy

Pre-clinical studies suggest that phage therapy targeting pks+ *E. coli* can reduce their oncogenic effects in vivo [136]. This approach might be applicable to humans, as early-phase clinical trials have shown the feasibility of using species-specific phage cocktails to target pathogens like Klebsiella pneumoniae in healthy patients [137].

### 5.4. Immunomodulator Therapies

Studies have demonstrated that *E. coli* strains can influence chemotherapy efficacy and resistance in CRC [138]. For instance, *E. coli Nissle 1917* expression of a long isoform of the enzyme cytidine deaminase (CDDL) conferred gemcitabine resistance in xenografts, while CDDL-deficient *E. coli* did not affect oxaliplatin resistance [139]. This resistance was attributed to bacterial metabolism of the drug, leading to its depletion. Additionally, pks+ *E. coli* have been found to translocate to mesenteric lymph nodes, reducing the cytotoxic T-cell abundance and inhibiting immune cell invasion in tumor tissue margins, thereby hindering the efficacy of anti-PD-1 treatment in CRC xenografts. These findings underscore the potential of specific bacterial strains, such as *E. coli*, to modulate the chemotherapy response and resistance in CRC patients.

CRC tumor tissues with colibactin-producing *E. coli* are associated with a decrease in tumor-infiltrating T lymphocytes (CD3 + T-cells) [140] and colibactin-producing *E. coli* was shown to decrease CD3+ and CD8+ T-cells and increase colonic inflammation in mice. A significant decrease in anti-tumor T-cells was observed in the mesenteric lymph nodes (MLN) of colibactin-producing *E. coli*-infected mice compared to the control and colibactin-producing *E.coli* infection reduces the efficacy of anti-mouse PD-1 immunotherapy in the MC38 tumor model [140], suggesting colibactin-producing *E. coli* can disrupt the antitumor T-cell response, leading to tumor resistance to immunotherapy [140]. Therefore, colibactin-producing *E. coli* could be a new biomarker that predicts the anti-PD-1 response in CRC, and it might be of interest to clear gut lumen from pks+ *E. coli* prior to providing immunotherapy in CRC patients.

*E. coli* vaccines against invasive diseases are a new way of enhancing the host immune response. In a study of the extraintestinal pathogenic *Escherichia coli* (ExPEC) utilizing pan-virulosome analysis of over 20,000 sequenced *E. coli* strains, the secreted cytolysin α-hemolysin (HlyA) was identified as a high-priority target for vaccine studies. Using the inactive purified form of HlyA via an autologous host secretion system has been shown to provide protection against several types of ExPEC infections, significantly reducing the bacterial load in many cases. The combination of an autotransporter (SinH) with HlyA was significantly effective and boosted the protection against a mixture of 10 highly virulent sequence variants and strains. These suggest bacterial products might be used as adjuvant therapy [141].

## 6. Screening and Early Detection

CRC diagnosis primarily relies on early detection through routine colonoscopy for asymptomatic patients and clinical presentation for symptomatic patients. Since CRC is often linked to changes in specific bacterial genera or species, diagnostic models incorporating microbial profiles, metabolites, or bacterial gene signatures could provide valuable tools for diagnosis or mass screening without invasive procedures [142].

### 6.1. Colorectal Cancer Screening

Routine screening for CRC and precancerous lesions remains essential for early detection and timely intervention. Screening modalities such as colonoscopy, fecal occult blood testing, and stool DNA testing can help identify individuals at increased risk of CRC, allowing for early detection of pks+ bacteria-associated malignancies. The choice of accurate methods (e.g., PCR, immunohistochemistry) to identify and quantify pks+ *E. coli* is the main challenge and such methods need to be validated in stool and biopsy samples, and the related costs need to be compared to existing CRC screening methods such as colonoscopy and fecal occult blood immunochemical tests.

### 6.2. Genetic Counseling and Testing

Genetic counseling involves evaluating a person’s personal and family medical history to determine the possibility of an inherited disease, such as FAP or Lynch syndrome, that is associated with an increased risk of developing CRC.

Now, in individuals with a family history of CRC syndromes associated with pks+ bacterial infections, genetic consulting should be integrated. At least in familial adenomatous polyposis (FAP) and in Lynch syndrome, identification of pks+ genetic disorders may be relevant [36,143]. Testing may include targeted analysis of genes known to be involved in these syndromes or a more extensive genomic analysis to identify potential mutations.

### 6.3. Evolution of Precision Medicine in CRC

Molecular alterations fostering the progression of CRC are acquired early in the carcinogenesis process, and there is inter-connectivity among genomic drivers (gene mutations and chromosomal instability), transcriptomic subtypes (microsatellite instability immune, canonical, metabolic or mesenchymal) and immune signatures (highly immunogenic, poorly immunogenic or inflamed and immune tolerant).

Emerging positive predictive markers for targeted therapies include infrequent genomic events, such as BRAFV600E mutations, ERBB2 amplifications, anaplastic lymphoma kinase (ALK) and neurotrophic receptor tyrosine kinase (NTRK) fusions and alterations in upstream nodes of the WNT pathway, such as ring finger protein 43 (RNF43), zinc and ring finger 3 (ZNRF3) and R-spondin (RSPO) genes. For immune checkpoint inhibitors, promising biomarkers include microsatellite instability and DNA polymerase-ε (POLE) mutations. Biomarker–drug co-development has evolved to accommodate a “multi-molecular, multi-drug” perspective of precision medicine. Combination therapies to halt tumor evolution and tackle minimal residual disease is in perspective.

## 7. Conclusions

The discovery of pks enzymes and their role in synthesizing various compounds has led to significant advancements in understanding microbial pathogenesis and its implications for human health. Among the compounds produced by pks, colibactin stands out due to its genotoxic effects and association with CRC. Colibactin, encoded by the pks gene, has been identified as a cyclomodulin that induces DNA damage, disrupts the eukaryotic cell cycle, and contributes to the initiation and progression of CRC. Strategies aimed at preventing pks+ occurrence, inhibiting colibactin production, enhancing host immune responses, promoting early detection, and advancing clinical management are crucial for mitigating the risk of CRC associated with pks+ bacteria. Developing effective interventions to improve the outcomes for pks+ *E. coli*-harboring CRC patients are in progress.

## Figures and Tables

**Table 1 microorganisms-12-01111-t001:** Overview of adherence factors and virulence mechanisms in pathogenic *E. coli* strains.

Pathotype	Adhesion Site	Virulence Factors	Mechanism/Effect	Disease Manifestations	References
Enteropathogenic *E. coli* (EPEC)	Small bowel enterocytes	Lymphostatin (LifA/Efa1)—EAF plasmid-encoding BFP and per locus—Type III secretion system and effectors (Tir, EspF, EspG, EspH, Map)—Enterotoxin (EspC)	Stimulates Cdc42-dependent filopodia formation, disrupts mitochondrial membrane potential	Diarrhea, intestinal inflammation, increased permeability, loss of absorptive surface area	[20,22,24,25]
Enterohaemorrhagic *E. coli* (EHEC)	Colon	Shiga toxin (Stx), Urease, Cif, —LEE pathogenicity island encoding, Type III secretion system and effectors (Tir, EspF, EspG, EspH, Map)—Enterotoxin (EspC), ToxB, Pet, and StcE, Lymphostatin (LifA/Efa1)	Cell lyse, cleaves ribosomal RNA, disrupting protein synthesis and killing cells, inducing renal damage, apoptosis, and local colonic damage	Bloody diarrhea (hemorrhagic colitis), non-bloody diarrhea, hemolytic uremic syndrome (HUS)	[20,22,24,26]
Enterotoxigenic *E. coli* (ETEC)	Small bowel enterocytes	Heat-stable enterotoxin a (STa), Heat-stable enterotoxin b (STb), Heat-labile enterotoxin (LT)	Increases cyclic AMP/GMP concentration, leading to ion secretion, Increases Ca^2+^ concentration, Colonization is mediated by colonization factors (CFs), and LT acts similarly to cholera toxin (CT)	Induces watery diarrhea	[20,22,27]
Enteroaggregative *E. coli* (EAEC)	Small and large bowel epithelia in a thick biofilm	Aggregative adherence fimbriae (AAFs), dispersin—Enterotoxins: Pic, ShET1, EAST1, Pet—Virulence factors regulated by AggR	Pathogenesis involves colonization of the intestinal mucosa, secretion of enterotoxins (e.g., EAST1), and induction of mucosal damage.	Causes persistent diarrhea in children and adults	[22,24,28,29,30]
Enteroinvasive *E. coli* (EIEC)	Colonic epithelial cells	Type III secretion system—Plasmid-encoded IcsA (outer-membrane protein), SepA (serine protease)—Virulence plasmid-encoded factors, VirA, IpaA, IpaB, IpaC, IpgD, IpaH	It invades epithelial cells, multiplies intracellularly, and induces apoptosis, Actin depolymerization, activation of Cdc42 and Rac	Causes inflammatory colitis similar to Shigella—Invasion, intracellular multiplication, and dissemination, Dysentery, watery diarrhea	[20,22,31,32]
Diffusely adherent *E. coli* (DAEC)	Epithelial cells diffusely	Adhesin: Dr family (F1845)—Cytotoxic necrotizing factor, autotransporter toxin Pet	Induces cytopathic effects and might impair brush-border enzymes	Implicated in diarrhea, particularly in children >12 months, cytopathic effects, impaired brush-border-associated enzymes	[22,24,33,34]
Uropathogenic *E. coli* (UPEC)	Uroepithelium	Specific adhesins: (P, type 1, F1C, S, M, Dr fimbriae) (Pap), — Toxins: Hemolysin, cytotoxic necrotizing factor (CNF-1, -2), Sat, HIyA	Altered cytoskeleton, necrosis, cell lysis	Uncomplicated cystitis, acute pyelonephritis	[20,22,24,34,35,36]

**Table 2 microorganisms-12-01111-t002:** Cyclomodulins of *E. coli*.

Cyclomodulins Produced by *E. coli*	Host Gene(s)	Mechanism	Reference
Cytotoxic necrotizing factor	cnf1	Modifies Rho GTPases, locking them in an active state, inducing cell cycle progression, leading to multinucleation and potentially contributing to carcinogenesis	[47]
Cytolethal-distending toxin (CDT)	cdtA, cdtB, and cdtC	Cell cycle arrest, DNA double-strand breaks, and chromosomal aberrations, potentially leading to senescence or apoptosis	[51]
Intimin-dependent attachment	eae, Type III secretion system	Effector eae downregulates the DNA mismatch repair system, resulting in DNA strand breaks	[49]
Cell cycle-inhibiting factor	cif	Irreversible cell cycle arrest at the G2/M transition through activation of a DNA damage-independent signaling pathway, inhibition of the ubiquitin/proteasome pathway, and accumulation of cyclin-dependent kinase inhibitors.	[48]
Colibactin	pks locus	Introduction of DNA strand breaks due to reactive cyclopropane group	[16]

**Table 3 microorganisms-12-01111-t003:** Functions of colibactin-associated genes.

Gene Name	Function	References
ClbA	Phosphopantetheinyl transferase for activation of megasynthases	[93]
ClbB	Biosynthesis of NRPS/pks hybrids	[94]
ClbC	Biosynthesis of pks	[94]
ClbD	Hydroxyl-acyl-CoA dehydrogenase	[95]
ClbE	Freestanding acyl carrier protein (ACP)	[95]
ClbF	α β dehydrogenase (αβdhg)	[96]
ClbG	Acyltransferase (AT)	[96]
ClbH	Cyclopropane-formatting synthetase	[92]
ClbI	Cyclopropane-formatting synthetase	[92]
ClbJ	Biosynthesis of NRPS	[96]
ClbK	Biosynthesis of NRPS/pks hybrids	[94]
ClbL	Amidase (role not fully known)	[96]
ClbM	Efflux pump for exporting pre-colibactin	[94]
ClbN	Initiates synthesis using asparagine to generate the prodrug motif N-myristoyl-d-asparagine (NMDA)	[92]
ClbO	Final enzymatic module of the assembly line and completes the extension stage of the pre-colibactin skeleton	[92]
ClbP	Periplasmic peptidase associated with bacterial cytoplasmic membrane	[96]
ClbQ	Thioesterase (TE)	[96]
ClbR	LuxR-type transcriptional activator	[92]
ClbS	Protects colibactin-producing bacterium from genotoxic effects	[96]

**Table 4 microorganisms-12-01111-t004:** Effects of colibactin.

Type of Effect	Description	References
Genotoxic	Colibactin alkylates DNA, leading to interstream crosslinks and DNA double-strand breaks, promoting genomic instability and mutations in critical oncogenes or tumor suppressor genes.	[112,116]
Genomic Instability	pks+ bacteria impair DNA repair mechanisms, increase mutagenic events, and promote tumor heterogeneity, facilitating tumor evolution and progression.	[117]
Inflammation	pks+ bacteria induce chronic inflammation in the colon, creating a pro-inflammatory microenvironment conducive to tumor development and progression.	[114]
Disruption of Gut Epithelial Barrier	pks+ bacteria disrupt the gut epithelial barrier, leading to increased permeability of the intestinal mucosa and leakage of bacterial toxins and antigens, contributing to carcinogenesis.	[120]
Modulation of Host Immune Responses	pks+ bacteria modulate host immune responses in the colon, promoting an immunosuppressive microenvironment favorable for tumor growth and progression.	[16]
Cell Cycle/Senescence	Colibactin-induced DNA damage triggers cell cycle arrest and senescence, affecting tumor cell proliferation and potentially influencing the anti- or pro-tumor role of senescence.	[115,121]
β-glucuronidase Activity	β-glucuronidase activity in *E. coli* releases carcinogenic methylazoxymethanol in the colon, with inhibitors reducing tumor formation in animal studies.	[125]

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
