# Peer review of "Contribution of pks+ *Escherichia coli* (*E*. *coli*) to Colon Carcinogenesis"

_microorganisms, 2024, doi:10.3390/microorganisms12061111_

Round 1
Reviewer 1 Report
Comments and Suggestions for Authors
In the manuscript submitted to me for review entitled "Contribution of E. Coli PkS+ in colon carcinogenesis“ the authors Mohammad Sadeghi, Denis Mestivier and Iradj Sobhani present an extensive review in which they analyze the role of the gut microbiota in colon carcinogenesis.
The submitted manuscript summarizes data published mainly in the last 3 decades. Various possible mechanisms that could influence the development of tumor disease to varying degrees have been considered. The authors elucidate how colibactin-producing bacteria induce DNA damage, disrupt the intestinal epithelial barrier, induce mucosal inflammation, modulate host immune responses, and influence cell cycle dynamics.
The systematization of such information may contribute to the encouragement of further research in this area by other researchers. This could lead to the development of mass screening of the gut microbiota and timely detection of bacterial species involved in the development of colorectal carcinoma (CRC). New approaches can also be developed to control and modulate the species composition of the gut microbiota to largely avoid the possibility of developing CRC.
In their study, the authors used 134 references, with nearly 1/4 of the total number being from the last 5 years. The fact that literary sources from recent years as well as research from 20-30 years ago are mentioned shows that the topic in the present manuscript has been considered for years. In my opinion, this topic will continue to be the subject of research until control of the gut microbiota is achieved, leading to improved human health. Therefore, I suggest that the present manuscript will attract the attention of readers of Microorganisms.
I did not notice any use of unnecessary self-citations, and all references provided information necessary to shape the manuscript. The data included is summarized with 3 extremely well-designed and presented figures and 4 tables, which I believe will be positively appreciated by readers. The authors' conclusions are consistent with the data presented in the manuscript.
My remarks and recommendations to the authors are:
1. I think it would look better if the references cited in Table 1 and 2 were centered as presented in Table 3.
2. Why do figures 1, 2 and 3 have two inscriptions each - above and below the figure? Why is this necessary?
3. After line 498 follows a table, but it is not numbered and titled. According to the order of the tables so far, this should be Table 4.
4. The Author Contributions and Conflicts of Interest sections are omitted in the back of the manuscript. Let the authors add them.
5. In the References section, reference number 128 does not indicate the year of publication. Let the authors add it.
Author Response
Response to Reviewer 1 Comments
Dear Reviewer,
First of all, we would like to thank you for your positive and thorough review of our manuscript entitled "Contribution of E. Coli PkS+ in Colon Carcinogenesis." We appreciate your constructive feedback and are pleased to address each of your recommendations.
Centering References in Tables 1 and 2 :
We have centered the references in Tables 1 and 2 as you suggested, ensuring consistency with Table 3.
Figure Captions :
We have removed the titles above Figures 1, 2, and 3. The captions now appear only below the figures, enhancing clarity and presentation.
Numbering and Titling Table 4 :
We have added the missing number and title for Table 4, following the sequence and format of the previous tables.
Author Contributions and Conflicts of Interest:
We have included the Author Contributions and Conflicts of Interest sections at the end of the manuscript as requested.
Reference Number 128 :
We have re-checked and updated reference number 128 to include the year of publication. The format provided by Zotero software has been adjusted accordingly.
We believe that these changes have improved the manuscript and we are grateful for your detailed review and valuable suggestions. Thank you once again for your positive feedback and for considering our manuscript for publication in Microorganisms.
Sincerely,
Mohammad Sadeghi, Denis Mestivier, and Iradj Sobhani

Reviewer 2 Report
Comments and Suggestions for Authors
Studying the relationship between gut microbes and colon cancer is a topic of interest to researchers. The present study reviewed the role of E. coli PkS+ in colon carcinogenesis, which mainly includes inducing DNA damage, promoting genomic instability, disrupting the gut epithelial barrier, inducing mucosal inflammation, regulating host immune responses and influencing cell cycle dynamics. influence cell cycle dynamics. I think this review can be accepted for publication in the journal of Microorganisms after minor revision.
such as:
1. line 24, It is suggested that the authors replace “Cancer; Colon; Microbiota” with “ Colorectal cancer; Gut microbiota”.
2. line 88, "Escherichia coli (E. coli)" , Escherichia coli is not the first occurrence in the text and does not need to be spelled out in full, in addition E. coli needs to be italicized.
Author Response
Response to Reviewer 2 Comments
Dear Reviewer,
Thank you very much for your insightful comments and positive feedback on our manuscript entitled "Contribution of E. Coli PkS+ in Colon Carcinogenesis." We are grateful for your suggestions and have made the necessary revisions as follows:
Line 24 :
As per your suggestion, we have replaced "Cancer ; Colon ; Microbiota" with "Colorectal cancer ; Gut microbiota."
Line 88 :
We have corrected the text to avoid repeating "Escherichia coli" in full since it is not the first occurrence. Additionally, we have ensured that "E. coli" is italicized.
We appreciate your constructive feedback, which has helped improve our manuscript. Thank you for considering our work for publication in Microorganisms.
Sincerely,
Mohammad Sadeghi, Denis Mestivier, and Iradj Sobhani
